# Predicting Failures of LLMs to Link Biomedical Ontology Terms to Identifiers: Evidence Across Models and Ontologies

Daniel B. Hier
*Department of Neurology and Rehabilitation*
*University of Illinois at Chicago*
Chicago, IL, USA
dhier@uic.edu

Steven K. Platt
*Lab for Applied Artificial Intelligence*
*Loyola University Chicago*
Chicago, IL, USA
splatt1@luc.edu

Tayo Obafemi-Ajayi
*Engineering Program*
*Missouri State University*
Springfield, MO, USA
tayoobafemiajayi@missouristate.edu

*Abstract*—**Large language models (LLMs) often perform well on biomedical NLP tasks but may fail to link ontology terms to their correct identifiers (IDs). We investigate why these failures occur by analyzing predictions across two major ontologies—Human Phenotype Ontology (HPO) and Gene Ontology–Cellular Component (GO-CC)—and two high-performing models, GPT-4o and LLaMa 3.1 405B. We evaluate nine candidate features related to term familiarity, identifier usage, morphology, and ontology structure. Univariate and multivariate analyses show that exposure to ontology identifiers is the strongest predictor of linking success. In contrast, features like term length or ontology depth contribute little. Two unexpected findings emerged: (1) large "ontology deserts" of unused terms predict near-certain failure, and (2) the presence of leading zeroes in identifiers strongly predicts success in HPO. These results show that LLM linking errors are systematic and driven by limited exposure rather than random variability. Encouraging consistent reporting of ontology terms paired with their identifiers in biomedical literature would reduce linking errors, improve normalization performance across ontologies such as HPO and GO, enhance annotation quality, and provide more reliable inputs for downstream classification and clinical decision-support systems.**

*Index Terms*—**ontology, normalization, large language models, Gene Ontology, Human Phenotype Ontology, logistic regression, biomedical term normalization**

## I. INTRODUCTION

Normalization of medical concepts to an ontology is a key aspect of the natural language processing of biomedical text. Biomedical knowledge embedded in free-text clinical notes must be transformed into a computable form to support clinical decision-making, population health management, biomedical research, and precision medicine [1]. Up to 80% of the information in electronic health records (EHRs) resides in unstructured text such as discharge summaries, radiology reports, progress notes, and consultation notes [2], [3]. Term normalization refers to the mapping of extracted clinical phrases to standardized ontology concepts and their identifiers. This is essential for enabling downstream natural language processing (NLP) applications, data analysis, and decision support [1], [4]. Biomedical term normalization typically consists of five sub-processes: (1) identifying terms of interest in free text, (2) extracting the terms, (3) assigning each term to a relevant ontology, (4) standardizing the term to a canonical ontology entry, and (5) linking the entry to its unique identifier (Table I) [5].

TABLE I
BIOMEDICAL TERM NORMALIZATION

| Step | Subprocess | Example |
|---|---|---|
| 1 | Identify target phrase | The patient was **ataxic**. |
| 2 | Extract target phrase | **ataxic** |
| 3 | Categorize | Use HPO |
| 4 | Standardize term | **Ataxia** |
| 5 | Link to ID | **HP:0001251** |

While early rule-based and traditional NLP systems showed only modest success in completing this process [6], [7], recent advances in large language models (LLMs) have produced dramatic gains in concept extraction and normalization performance [8]–[14]. However, reported performance may overstate effectiveness for two reasons: (1) benchmark datasets are often biased toward common or easily normalized terms [15]–[17], and (2) evaluation metrics typically reflect aggregate performance across the entire normalization pipeline, obscuring specific sources of failure such as identifier linking errors [13]. The final step—*linking an ontology entry to its correct ontology identifier*—is particularly vulnerable to failure.

Although these failures may appear random, emerging evidence suggests they result from systematic imbalances and deficiencies in the training corpora \cite{do2024mapping}. Building on this insight, we evaluate a broad set of linguistic, structural, and usage-based features to better understand and predict failures in normalizing biomedical ontology terms. Using predictions from two state-of-the-art LLMs (LLaMa 3.1 405B and GPT-4o), we analyze 18,988 Human Phenotype Ontology (HPO) terms and 4,023 Gene Ontology (GO) cellular component terms. We engineer candidate predictors across three dimensions: (1) corpus-based frequency features (e.g., PubMed Central (PMC) identifier counts, UniProtKB annotation prevalence), (2) morphology-based features (e.g., term length, entropy, digit prefixes), and (3) ontology structure-

based features (e.g., depth, leaf status). Using univariate statistics and logistic regression, we quantify each feature's contribution to linking accuracy. Our results show that term and identifier usage patterns in the literature are more predictive of linking performance than intrinsic structural properties of the terms or the ontology.

The primary contributions of this work are:

1) We introduce a comprehensive set of term-, identifier-, and ontology-level features to predict failures of large language models (LLMs) in ontology ID linking.

2) We show that features capturing identifier familiarity (`PMC Identifiers`, `No Annotation`, and `Annotation Count`) are the strongest predictors of successful term-to-identifier linking.

3) We identify problematic terms that cluster in *ontology deserts*—regions of sparse or unused terms—which account for a substantial fraction of normalization errors.

4) We uncover a novel formatting artifact, the *leading zero effect*, illustrating how LLMs exploit tokenization patterns to improve linking accuracy.

5) We demonstrate that these findings generalize across two major biomedical ontologies (HPO and GO) and two high-performing LLMs (GPT-4o and LLaMa 3.1 405B), providing actionable insights for improving ontology annotation practices in biomedical literature. While Gene Ontology annotation has gained widespread adoption, HPO annotations in the biomedical literature have lagged behind (Table III).

## II. METHODS

*Data*.

We analyzed 18,988 terms from the Human Phenotype Ontology (HPO) and 4,023 terms from the Gene Ontology's Cellular Component (GO-CC) hierarchy (Table III). Ontologies were downloaded from the National Center for Biomedical Ontology (NCBO) BioPortal in CSV and OBO formats [18].

*Language Models Evaluated*.

To evaluate the capacity of large language models (LLMs) to accurately link ontology terms with their corresponding identifiers, we employed the open-access `Meta-Llama-3.1-405B-Instruct-Turbo` model via the Together AI platform (https://www.together.ai/) and OpenAI's `GPT-4o` accessed via https://platform.openai.com/. The query prompt for both platforms was the same:

```
prompt = "What is the HPO ID
    for {'hpo_term'}?
Return only the code in
    format HP:1234567"
```

*Feature Construction*.

We constructed nine features, grouped into five conceptual categories, to investigate what properties predicted success in linking a biomedical term to its ID. These are summarized in Table II. Ontology graphs (HPO and GO) were parsed from OBO files using the *obonet* Python package and converted to directed acyclic graphs using NetworkX, retaining

only `is_a` relationships. Additional graph-derived features such as *In_Degree*, *Ancestor_Count*, and *Subgraph_Size* were explored, but only *Leaf* and *Depth* were retained for modeling.

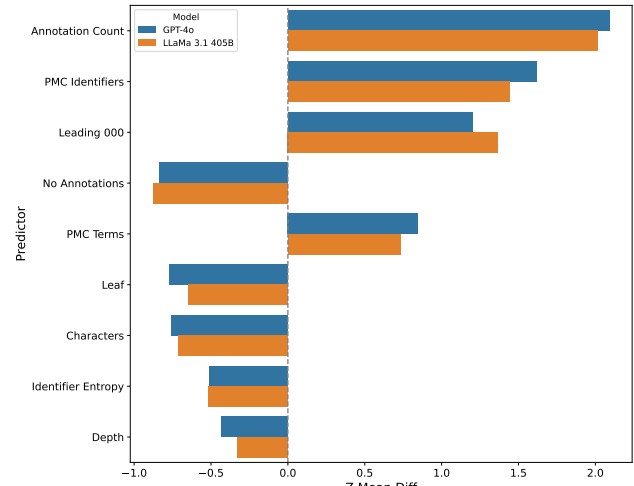

Fig. 1. **Univariate predictors of successful term-to-identifier linking for HPO.** Bars show the difference in mean standardized (z-score) feature values between correctly and incorrectly linked terms. Features are sorted by absolute effect size, with the largest differences shown at the top. Positive bars (e.g., `Annotation Count`) indicate higher values for correctly linked terms, while negative bars (e.g., `No Annotations`) indicate higher values for incorrect links. Results are shown for GPT-4o (blue) and LLaMa 3.1 405B (orange), which demonstrate broadly similar patterns.

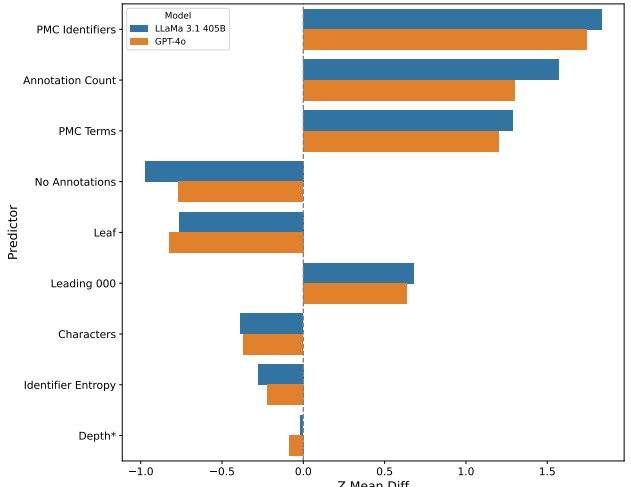

Fig. 2. **Univariate predictors of successful term-to-identifier linking for GO-CC.** Bars show the difference in mean standardized (z-score) feature values between correctly and incorrectly linked terms. Features are sorted by absolute effect size, with the largest differences shown at the top. Positive bars (e.g., `Annotation Count`) indicate higher values for correctly linked terms, while negative bars (e.g., `No Annotations`) indicate higher values for incorrect links. Results are shown for GPT-4o (blue) and LLaMa 3.1 405B (orange), which demonstrate broadly similar patterns.

Familiarity metrics were obtained from PubMed Central (PMC) [19] using the NCBI E-Utilities API. Frequency counts were gathered by issuing programmatic queries

to the PMC search endpoint and parsing total hits from JSON responses. To avoid rate limits, requests were throttled with a one-second delay. Disease-to-HPO annotations (`phenotype.hpoa`, 272,060 entries) were obtained from https://hpo.jax.org/data/annotations, and protein-to-GO annotations (`uniprot_sprot.dat`, 842,424 entries) from https://www.uniprot.org/help/downloads.

`ID Entropy` was computed as Shannon entropy (Equation 1):

$$H = -\sum p_i \log_2 p_i \qquad (1)$$

where $p_i$ is the frequency of character $i$ in the ID string.

TABLE II
CANDIDATE PREDICTIVE FEATURES

| Category | Feature Description |
|---|---|
| **Term Familiarity** | |
| PMC Terms | Count of occurrences of term in PubMed Central (PMC) |
| **Identifier Familiarity** | |
| PMC Identifiers | Count of occurrences of ontology Identifier in PMC |
| No Annotation | Binary flag indicating whether an ontology term was used to annotate a disease (HPO) or a protein (GO) |
| Annotation Count | Count of times each term was used to annotate a protein (GO) or a disease (HPO) |
| **Term Morphology** | |
| Characters | Character length of the term |
| **Identifier Morphology** | |
| Leading 000 | Binary flag for Identifiers where the first three digits of the 7-digit string are **000** |
| Identifier Entropy | Shannon entropy of the Identifier string |
| **Graph Metrics** | |
| Leaf | Binary flag for terms that are a leaf node in the ontology |
| Depth | Distance of term to the ontology root node |

*Zipf Plots.*
Zipf [20] observed that the use of words in natural language follows a power law distribution so that a few words are used frequently, while many are used rarely. Zipf ranked vocabulary terms by their usage frequency (with rank 1 being most frequent) and plotted the base-10 logarithm of term frequency (y-axis) against the base-10 logarithm of rank (x-axis). The result was a straight line with a negative slope, now known as a *Zipf plot*. The usage of terms in biomedical vocabularies such as HPO and GO follows a Zipfian distribution, with a small number of highly annotated terms and a large number of sparsely or unannotated terms. To visualize this pattern, we constructed Zipf plots for the HPO and GO ontologies. Terms were ranked by their disease annotation counts (HPO) or protein annotation counts (GO), from highest to lowest frequency. HPO terms were ranked from 1 to 18,988, and GO terms from 1 to 4,023. Since many terms have zero

annotations and to avoid computing $\log_{10}(0)$, 0.1 was added to all annotation counts.

We plotted $\log_{10}(\text{rank})$ versus $\log_{10}(\text{count})$ for 2,000 randomly sampled terms from each ontology. Terms were color-coded according to annotation status: red for terms with no annotations, green for terms correctly linked to their ontology ID, and blue for terms incorrectly linked. To reduce overplotting in the densely packed long tail of the distribution, jitter was applied to the marker positions. The resulting plots are shown for HPO (Fig. 7) and GO (Fig. 8). Since there is increasing evidence that LLMs struggle to retrieve information from the long tail of frequency distributions [21], [22], we specifically constructed these Zipf plots to evaluate the performance of GPT-4o and LLaMA 3.1 405B on the *long tails* of the HPO and GO ontologies.

*Assessment of Feature Importance by Univariate Analysis.*
We evaluated the predictive value of nine candidate features in determining whether a large language model (LLM) would successfully normalize an ontology term to its correct identifier. For each feature, we calculated the mean z-score separately for correctly linked and incorrectly linked terms (Figs. 1 and 2). Statistical significance was assessed using independent-sample *t*-tests, and effect sizes were estimated with Cohen's *d*. Figures display the difference between the mean z-scores for correctly and incorrectly linked terms, with features ranked by the absolute magnitude of *d* and grouped by direction to indicate whether they were positively or negatively associated with linking success.

*Assessment of Feature Importance by Logistic Regression.*
We fitted separate logistic regression models for the HPO and GO-CC ontologies using nine standardized predictors to estimate the probability of successful term-to-identifier linking. Models were trained and evaluated for both GPT-4o and LLaMA 3.1 405B using the *statsmodels* library, with predictors standardized via *StandardScaler* (mean = 0, SD = 1). Feature importance was assessed by the magnitude and direction of standardized regression coefficients (Figs. 3 and 4). To validate these coefficients, we computed SHAP values and confirmed consistent magnitude and direction (data not shown). Model performance was evaluated using accuracy, precision, recall, F1 score, McFadden's $R^2$, and Tjur's $R^2$ (Table IV).

## III. RESULTS

Although the Human Phenotype Ontology (HPO) and the Gene Ontology – Cellular Component (GO-CC) share a foundational design philosophy grounded in OBO principles [23], they differ substantially in structure, scale, and usage patterns (Table III). GO-CC contains fewer total concepts, a shallower hierarchy, a higher proportion of leaf nodes, and significantly more annotations per term—reflecting its emphasis on protein localization. In contrast, HPO exhibits a deeper structure, longer average term length, and a greater prevalence of identifier formatting artifacts such as the `Leading 000` prefix.

A shared property of both ontologies is the large proportion of unused terms: over 50% in GO-CC and over 40% in

TABLE III
CELLULAR COMPONENT HIERARCHY OF GENE ONTOLOGY (GO-CC)
COMPARED TO HUMAN PHENOTYPE ONTOLOGY (HPO)

| Metric | Type | GO-CC | HPO |
|---|---|---|---|
| Concepts | count | 4,023 | 18,988 |
| Accuracy (LLaMa 3.1 405B) | % | 16.7 | 9.3 |
| Accuracy (GPT-4o) | % | 17.8 | 8.8 |
| Unigram Terms | % | 5.6 | 7.3 |
| Leading 000 Identifiers | % | 15.9 | 34.0 |
| Unused Identifiers | % | 54.4 | 40.4 |
| Terms that are a Leaf | % | 78.8 | 69.3 |
| Hierarchy Depth | mean | 4.0 | 6.9 |
| Identifier Count in PMC | mean | 9.9 | 1.1 |
| Term Count in PMC | mean | 3,962 | 30,903 |
| Length of term | mean | 26.8 | 32.6 |
| Annotations per Term | mean | 209.4 | 14.1 |
| Identifier Entropy | mean | 2.8 | 2.7 |

Accuracy = percentage of ontology terms correctly linked to their identifiers.

Unigram Terms = percentage of single-word terms in the ontology.

For other definitions, see Table II.

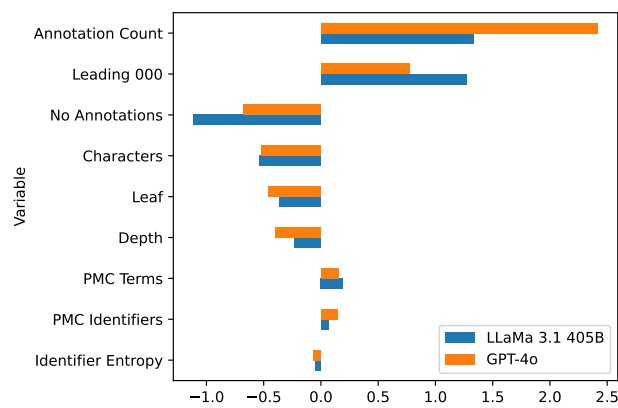

Fig. 3. **Standardized logistic regression coefficients for model to predict successful linking of HPO terms to their identifiers for LLaMa 3.1 405B and GPT-4o.** Both models showed a similar pattern with `Annotation Count` and `Leading 000` as the largest positive coefficients and `No Annotations` as the largest negative coefficient in the models.

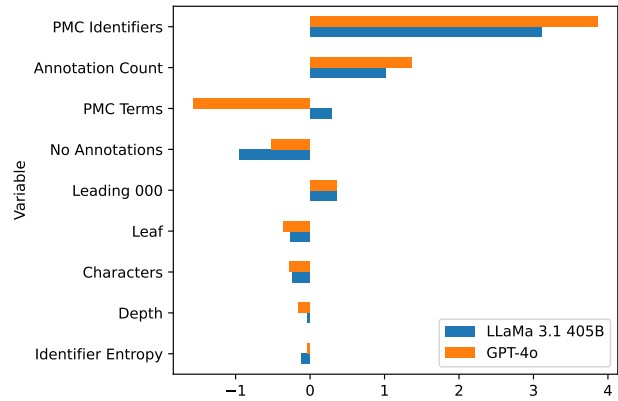

Fig. 4. **Standardized logistic regression coefficients for model to predict linking of GO term to GO identifiers.** Coefficents are ranked from largest to smallest. LLaMa 3.1 405B model and GPT-4o model had a similar pattern with `PMC Indentifiers` and `Annotation Count` having the biggest coefficients.

HPO, highlighting sparsely annotated regions we refer to as *ontology deserts*. These are terms that have never been used in practice—either to annotate a disease (HPO) or a protein (GO)—and may be underrepresented in LLM training corpora.

These structural and usage differences likely contribute to the disparity in raw linking accuracy across ontologies. For both LLMs evaluated—LLaMa 3.1 405B and GPT-4o—GO-CC accuracy was approximately double that of HPO. Specifically, LLaMa 3.1 achieved 16.7% accuracy on GO-CC and 9.3% on HPO, while GPT-4o reached 17.8% on GO-CC and 8.8% on HPO. This pattern likely reflects the much greater prevalence of GO term–identifier pairs in biomedical literature and protein annotation databases: on average, a GO-CC term is associated with 209.4 protein annotations, compared to just 14.1 disease annotations per HPO term (Table III). Greater training-time exposure increases the likelihood that a model will successfully learn and apply the correct term–ID mapping.

We began by evaluating each predictive feature independently to assess its association with successful linking (Figs. 1 and 2). Across both ontologies and models, familiarity signals—particularly `PMC Identifiers` and `Annotation Count`—were strongly and positively associated with linking accuracy. Conversely, the absence of annotations (`No Annotations`) emerged as a robust negative predictor, underscoring the importance of representation in curated datasets.

In contrast, intrinsic properties of the identifier—such as the `Leading 000` formatting artifact—were predictive only in HPO, likely reflecting ontology-specific ID conventions. Morphological features (e.g., `Characters`) and ontology topology metrics (e.g., `Depth`, `Leaf`) showed weaker and less consistent associations. These findings suggest that LLM linking success is driven more by exposure frequency than by structural or morphological complexity. Notably, this pattern generalized across two ontologies and two high-performing

LLMs, highlighting the central role of usage-based familiarity in term normalization.

To assess the independent contribution of each feature, we trained logistic regression models using the full feature set (Figs. 3 and 4). Both models performed well, achieving AUC values and classification accuracies above 0.88 (Table IV). The strongest predictors were `Annotation Count`, `PMC Identifiers`, `No Annotations`, and `Leading 000`. Notably, the `Leading 000` artifact had a strong positive effect in HPO but was negligible in GO, again pointing to differences in identifier conventions across ontologies. In contrast, structural and morphological features—including `Depth`, `Leaf`, `Characters`, and `Identifier Entropy`—made weaker or inconsistent contributions.

The relatively lower ranking of `PMC Identifiers` in the HPO model likely reflects its moderate correlation with `Annotation Count` ($r = 0.56$), which may reduce its

independent contribution when both variables are included in the regression. Together, these results reinforce the conclusion that exposure to term–identifier pairs in literature and curated datasets—not internal structure—is the dominant driver of LLM linking success.

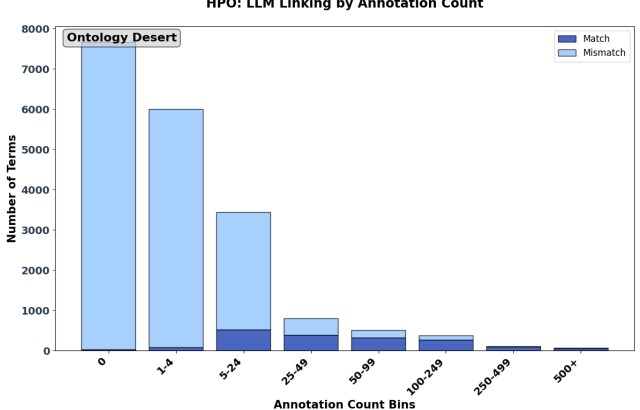

Fig. 5. **Annotation count predicts linking success for HPO terms.** Results shown for LLaMa 3.1 405B. Terms with no annotations (bin 0) have the highest failure rates. Note the distribution of annotations for terms is Zipfian with many terms with few counts (bin 0) and few terms with many counts (see bins to the far right). Dark blue shading shows terms correctly linked to ID, light blue shading shows failed linkings.

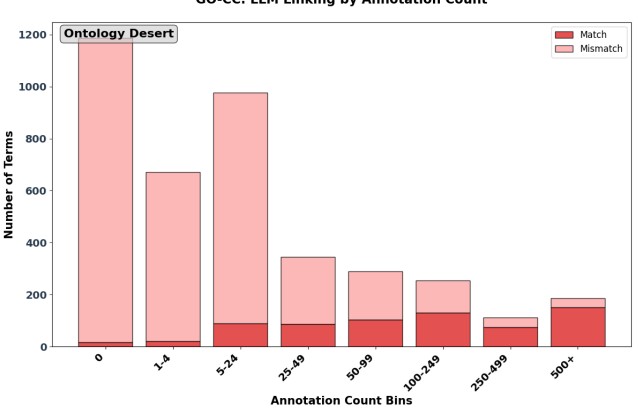

Fig. 6. **Annotation count predicts linking success for GO terms**. Results are shown for LLaMa 3.1 405B. Terms with no annotations (bin 0) have the highest failure rates. Note the distribution of annotations for terms is Zipfian with many terms with few counts (bin 0) and few terms with many counts (see bins to the far right) . Light red shading shows failed linking. Dark red shading shows correctly linked terms. Importantly, as annotation count increases so does linking accuracy.

## IV. DISCUSSION AND CONCLUSIONS

Our analysis demonstrates that failures in linking ontology terms to their correct identifiers by large language models (LLMs) such as LLaMa 3.1 405B and GPT-4o are not random. Instead, they are strongly predicted by features that reflect term familiarity—most notably, usage frequency in the biomedical literature and curated annotations. This was evident both when terms were binned by annotation count (Figs.5 and6) and

TABLE IV
PERFORMANCE OF LOGISTIC REGRESSION MODELS: PREDICTING
CORRECT LINKING OF TERM-TO-IDENTIFIER FOR HPO AND GO

| Ontology | LLM | Acc. | Prec. | Recall | F1 | AUC | MP-$R^2$ | Tjur's $R^2$ |
|----------|-----|------|-------|--------|-----|------|----------|--------------|
| HPO | LLaMa 3.1 | 0.93 | 0.75 | 0.39 | 0.51 | 0.94 | 0.41 | 0.46 |
| HPO | GPT-4o | 0.95 | 0.82 | 0.48 | 0.61 | 0.95 | 0.47 | 0.49 |
| GO | LLaMa 3.1 | 0.89 | 0.81 | 0.44 | 0.57 | 0.91 | 0.38 | 0.41 |
| GO | GPT-4o | 0.89 | 0.84 | 0.44 | 0.58 | 0.88 | 0.39 | 0.34 |

Accuracy (Acc.) = proportion of correctly classified term-to-ID predictions.
Precision (Prec.) = fraction of predicted correct matches that are truly correct.
Recall = fraction of true matches correctly identified by the model.
F1 = harmonic mean of precision and recall.
AUC = area under the ROC curve, measuring discriminative ability.
McFadden's Pseudo-$R^2$ (MP-$R^2$) = model fit statistic for logistic regression.
Tjur's $R^2$ = difference in average predicted probabilities between correct and incorrect matches.

when rank–frequency relationships were visualized using Zipf plots (Figs.7 and8). These insights were further confirmed by assessments of feature importance using both univariate and multivariate logistic regression analyses, which consistently identified annotation frequency and identifier exposure as dominant predictors. In contrast, morphological and structural features provided limited predictive value, indicating that linking success is primarily driven by term and identifier prevalence in biomedical corpora.

A striking insight is the prevalence of underutilized terms: 40.4% of HPO terms lack disease annotations, and 54.4% of GO-CC terms lack protein annotations (Table III). These unannotated terms define vast *ontology deserts*, where LLMs fail to correctly link terms due to lack of exposure during training. In our dataset, only 0.3% of unannotated HPO terms and 0.4% of unannotated GO terms were correctly matched, compared to substantially higher rates among annotated terms.

The predictive power of features such as `Annotation Count`, `No Annotations`, and `PMC Identifiers` aligns with the statistical nature of LLM training. Rather than inferring meaning from ontology structure or logical principles, LLMs rely on co-occurrence patterns in text corpora. Term–ID pairs seen more frequently are more likely to be linked correctly.

While LLMs can infer meanings of even rare biomedical terms from context, accurate ID normalization still depends on explicit exposure to the identifiers themselves. This reveals a core limitation of token-based pretraining: semantic understanding alone does not guarantee correct term-to-ID linkage.

The `Leading 000` feature emerged as an unexpectedly strong predictor for HPO terms, where terms beginning with `000` were linked more accurately. This may reflect consistent subtokenization patterns by the LLaMa 3 and GPT-4 tokenizers, which split HPO identifiers into predictable chunks. Recent work on tokenizer adaptation [24] emphasizes that such effects can shape downstream model behavior.

Based on these findings, we propose several strategies to improve LLM performance on ontology normalization tasks:
1) **Knowledge-based fine-tuning:** Supervised fine-tuning on curated term–identifier pairs can strengthen mappings

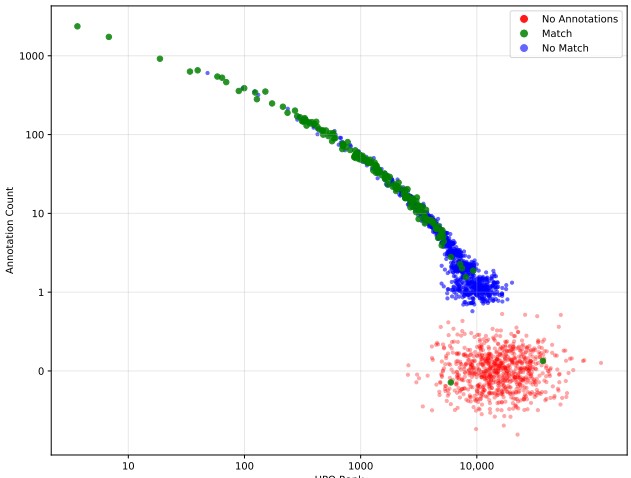

**Fig. 7. Zipf plot for HPO terms.** Human Phenotype Ontology (HPO) terms are ranked by the number of disease annotations. The plot shows log-transformed rank (x-axis) versus log-transformed annotation count (y-axis). Correctly linked terms are shown in green, incorrectly linked terms in blue, and terms with no annotations in red. Correct matches cluster in the high-frequency head of the distribution, while unannotated terms dominate the long tail. Markers are jittered to reduce overlap, and 2,000 randomly selected terms are displayed to improve readability.

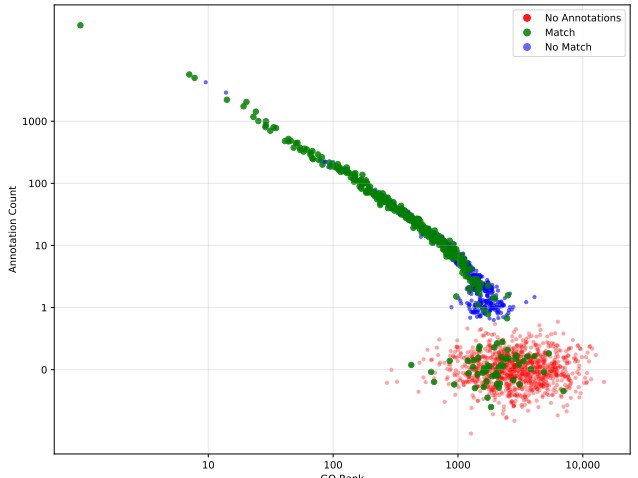

**Fig. 8. Zipf plot for GO-CC terms.** Gene Ontology – Cellular Component (GO-CC) terms are ranked by the number of protein annotations. The plot shows log-transformed rank (x-axis) versus log-transformed annotation count (y-axis). Correctly linked terms are shown in green, incorrectly linked terms in blue, and terms with no annotations in red. Correct matches cluster in the high-frequency head of the distribution, while unannotated terms dominate the long tail. Markers are jittered to reduce overlap, and 2,000 randomly selected terms are displayed to improve readability.

that are weakly encoded during pretraining [25].

2) **Retrieval-augmented generation (RAG):** Incorporating RAG architectures allows relevant term–identifier pairs to be dynamically retrieved and injected into the context window at inference time [26]–[28].

3) **Corpus-level interventions:** Improving how ontology terms are reported in biomedical literature may have the greatest long-term impact [29]–[31]. Authors should be encouraged to routinely pair ontology terms with their standard identifiers. Sundaramurthi et al. [32] exemplify best practices by aligning clinical phenotypes with HPO identifiers in a structured format. While the Gene Ontology (GO) has achieved widespread adoption—with a mean of 209.4 protein annotations per term—the Human Phenotype Ontology (HPO) lags, averaging only 14.1 disease annotations per term (Table III). This disparity highlights that, unlike GO, systematic annotation of every phenotype feature in the biomedical literature, as envisioned by Robinson and colleagues [29], has not yet been realized.

This study evaluated two ontologies (HPO and GO-CC) across two models (GPT-4o and LLaMa 3.1 405B) and is, to our knowledge, the first to systematically examine why LLMs fail at term-to-ID linking. Understanding these failures supports both model improvement and interpretability.

Future work will assess generalizability to other ontology branches (e.g., GO Molecular Function or Biological Process) and terminologies such as SNOMED CT, RxNorm, and LOINC. Notably, RxNorm and LOINC lack full ontological structure, which may affect performance. We also plan to evaluate alternative models such as Claude and Mistral, which differ in architecture and training corpora. Our current analysis focuses on exact matches; we aim to develop scoring metrics that capture partial subtoken overlap and semantic proximity.

In sum, successful term-to-identifier linking by LLMs depends on sufficient exposure to both the ontology term and its identifier. When either—particularly the identifier—is underrepresented, models often fail even when they accurately understand the concept. Encouraging consistent reporting of ontology terms paired with their identifiers in biomedical literature would reduce ontology deserts, improve normalization performance across ontologies such as HPO and GO, enhance annotation quality, and provide more reliable inputs for downstream classification and clinical decision-support tools.

## Acknowledgement

This work was supported in part by the National Science Foundation, Award Number 2423235.

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
