# OpenReview forum: "Predicting Failures of LLMs to Link Biomedical Ontology Terms to Identifiers: Evidence Across Models and Ontologies"
_IEEE.org/EMBS/BHI/2025/Conference — BHI 2025_

### Official Review · Reviewer_nK79 · 2025-07-15
**Overall, the paper is well-written. However, in the methodology section, some of the evaluation terms could be explained in greater detail to ensure the results section is easier to interpret and fully understood by the reader.**

**Confidence:** 5
**Clarity Of Writing:** good
**Clinical Significance:** fair
**Methodological Novelty:** good
**Overall Rating:** 6
**Final Rating:** 7

**Experiments And Results:**

good

**Questions For The Authors:**

1. It would be helpful if Table III included a brief explanation of the metrics being observed, along with their significance. This would provide readers with a clearer understanding of how these metrics contribute to evaluating the model’s performance.
2. After Table IV, in the paragraph, 'Unused terms refers to terms that...' should be corrected to ''Unused terms refer to terms that...'

**Strengths:**

1. The paper compares the LLM s efficiency with multiple evaluation metrics making the analysis more thorough
2. Comparing with multiple datasets makes the method more robust
3. Provides insight on what can be done in future research while developing such LLM model

**Summary Of The Paper:**

The paper explores the use of Large Language Models (LLMs) for explaining ontology terms, aiming to improve communication between doctors and patients. This application is particularly valuable, as interpreting medical terminology requires more nuanced reasoning compared to general language tasks. The discussion on the limitations of LLMs in this context adds depth to the study and highlights important areas for future improvement.

**Weaknesses:**

1. The 'Feature Extraction' and 'Results' sections would benefit from more detailed explanations, as some parts appear abrupt and could be expanded for better clarity and coherence.
2. The results are primarily presented using the LLaMA model. It would be interesting to see a comparison with other LLMs as well, which could offer additional insights into the generalizability and performance of the proposed approach.

---

### Official Review · Reviewer_mC3x · 2025-07-16
**Linking ontology to ID using LLM**

**Confidence:** 3
**Clarity Of Writing:** great
**Clinical Significance:** great
**Methodological Novelty:** good
**Overall Rating:** 6
**Final Rating:** 7

**Experiments And Results:**

good

**Questions For The Authors:**

The use of SHAP requires further clarification. Since logistic regression provide coefficients that reflect feature importance, it would be helpful for the authors to explain the specific need for SHAP analysis in this context. What additional insight does SHAP provide beyond the model coefficients?

**Strengths:**

1. The inclusion of a Zipf plot provides novel and insightful analysis of the data.
2. The paper is well written and easy to follow.
3. The topic is highly relevant and timely, given the growing integration of LLMs in biomedical informatics.

**Summary Of The Paper:**

The manuscript investigates the ability of large language models (LLMs) to link ontology terms to their corresponding IDs. The authors evaluate the LLaMA model using HPO and GO datasets and demonstrate limitations in this capability.

**Weaknesses:**

The study only evaluates the LLaMA model. While LLaMA may be a state-of-the-art model, testing additional models would greatly strengthen the conclusions. Alternatively, if testing other models is not feasible, the authors should narrow the scope of their claims to explicitly focus on LLaMA rather than generalizing to all LLMs. Focusing only on one model reduces the impact factor of the paper.

---

### Official Review · Reviewer_6oJQ · 2025-07-19
**Insightful analysis of LLM failures in ontology linking with strong feature modeling, but broader model comparisons and practical implications remain limited.**

**Confidence:** 3
**Clarity Of Writing:** good
**Clinical Significance:** poor
**Methodological Novelty:** fair
**Overall Rating:** 4
**Final Rating:** 6

**Experiments And Results:**

great

**Questions For The Authors:**

1. How did you select the ten features for your logistic regression? For example, in the case of ontology graphs, why did you include only leaf and depth while excluding other features like degree, ancestor count, or subgraph size? Are there other potential variables that might influence the prediction?

2. Your findings highlight key predictors of LLM failures in ontology ID linking. Beyond improving model performance, how might these results be applied in practice? Could you discuss the potential practical impacts of applying these insights more broadly?

3. The comparison and evaluation of other LLMs are necessary.

4. In TABLE V, you report the difference in logistic regression coefficients (Δ) between the HPO and GO models. Could you clarify the motivation behind including this comparison? How should we interpret these differences in the context of ontology-specific behavior?

**Strengths:**

1. The paper offers a methodologically rigorous breakdown of the factors influencing LLM success and failure in ontology ID linking, using both univariate and multivariate (logistic regression and SHAP) approaches.

2. The discovery of formatting artifacts (like the leading 000 effect) and the identification of "ontology deserts" bring novel perspectives to the ontology normalization problem.

**Summary Of The Paper:**

This paper investigates the reasons why large language models, despite strong biomedical NLP performance, often fail to link ontology terms to their correct IDs. Using data from the Human Phenotype Ontology (HPO) and the Gene Ontology Cellular Component (GO-CC), the authors evaluate ten features to identify key predictors of LLM failure. They find that the exposure frequency of ontology IDs is an important feature of linking accuracy, with surprising effects from features like leading zeroes in IDs and the presence of large "ontology deserts" of unused terms.

**Weaknesses:**

1. The study evaluates only one language model (LLaMA 3.1 405B), which may limit the general applicability of its findings across diverse architectures such as GPT-4, Claude, or Mistral.  The comparisons with other LLMs could help determine whether the observed linking failures are model-specific or indicative of a broader limitation across architectures.

2. The evaluation relies solely on exact ID matching, which may underestimate performance by ignoring semantically or hierarchically related predictions.

---

### Official Review · Reviewer_1UE6 · 2025-07-21
**Predicting the Failures of Large Language Models to Link Ontology Terms to Their IDs**

**Confidence:** 4
**Clarity Of Writing:** good
**Clinical Significance:** fair
**Methodological Novelty:** good
**Overall Rating:** 5

**Experiments And Results:**

good

**Questions For The Authors:**

One area I might be misunderstanding is how much of the observed linking failure is due to the LLaMA model itself versus the general nature of corpus-trained LLMs. Would results be expected to differ meaningfully with models trained differently (via retrieval or with knowledge graphs)? Clarifying this could affect the confidence in how broadly the findings could apply.

Relatedly, it wasn’t fully clear how the model was prompted. Understanding the prompting setup could help readers better judge the fairness and rigor of the evaluation.

Lastly, although the discussion touches on practical recommendations, a more explicit “translational statement” (how this work could directly improve clinical NLP tools or annotation pipelines) would strengthen the paper. If the authors clarified these real-world implications more clearly, it would likely raise my score on clinical significance.

**Strengths:**

One of the key strengths of this work is its clear and data-driven approach to clarify an underexplored problem in biomedical NLP. By systematically analyzing a wide range of features including term frequency, ontology structure, formatting quirks, the paper convincingly shows that LLMs rely heavily on exposure rather than understanding. The use of SHAP values and logistic regression to interpret model behavior is a nice touch, helping ground the findings in interpretable metrics. This work also does a great job of priming the field for next steps, like developing targeted fine-tuning strategies or using retrieval-augmented methods to patch gaps in LLM training. It encourages the community to think critically about how we curate training data and how we present ontology terms in literature.

**Summary Of The Paper:**

This paper attempts to understand why large language models often fail to link biomedical ontology terms to their correct IDs. The authors analyze term linking performance across HPO and GO-CC and find that the strongest predictor of success is how often an ID appears in literature/dataset and not anything structural about the ontology. They also uncovered a surprising finding of the positive effect of leading zeroes in IDs. Overall, the study makes a strong case that these failures are systematic and tied to gaps in training data, not model limitations.

**Weaknesses:**

A main weakness of the paper is that it has a limited focus on just one language model and only two biomedical ontologies, limiting generalization of the findings. It’s also worth noting that the evaluation focuses strictly on exact ID matching, without considering near misses or semantically related alternatives which might underestimate the model's actual understanding. While the analysis is thorough, the practical implications (applying RAG or fine-tuning in real-world systems, for example) could be better fleshed out more to help guide implementation.

---

### Official Review · Reviewer_CrRE · 2025-07-22
**Predicting the Failures of Large Language Models to Link Ontology Terms to their IDs**

**Confidence:** 5
**Clarity Of Writing:** good
**Clinical Significance:** good
**Methodological Novelty:** good
**Overall Rating:** 5

**Experiments And Results:**

good

**Questions For The Authors:**

Better emphase innovative part.
Better describe clinical beneftis.

**Strengths:**

Two ontologies are described in this manuscrikpt Human Phenotype Ontology and the Cellular Component branch of the Gene
Ontology —and evaluated only a single large language model: LLaMA 3.1 with 405 billion parameters.
Focus is attempt to understand why LLMs fail to predict identifiers of ontology terms.
They found that successful linking of an ontology term to its correct identifier by a LLM depends critically on sufficient exposure to both the surface form of the term and its corresponding ID during training.

**Summary Of The Paper:**

Two ontologies are described in this manuscrikpt Human Phenotype Ontology and the Cellular Component branch of the Gene
Ontology —and evaluated only a single large language model: LLaMA 3.1 with 405 billion parameters.
Focus is attempt to understand why LLMs fail to predict identifiers of ontology terms.
They found that successful linking of an ontology term to its correct identifier by a LLM depends critically on sufficient exposure to both the surface form of the term and its corresponding ID during training.
Better emphase innovative part.
Better describe clinical beneftis.

**Weaknesses:**

Better emphase innovative part.
Better describe clinical beneftis.